# Relics of War: Damaged Structures and Their Replacement or Management in Modern Landscapes

Peter J. Larkham [1,*] and David Adams [2]

1 Department of the Built Environment, Birmingham City University, City Centre Campus, Birmingham B4 7BD, UK
2 School of Geography, Earth and Environmental Science, University of Birmingham, Edgbaston, Birmingham B15 2TT, UK
* Correspondence: peter.larkham@bcu.ac.uk

**Abstract:** All wars, large and small scale, have had impacts on the built environments enmeshed in the conflict. This is almost always an adverse reaction, often involving destruction, but can also include the construction of defensive or other features. Damaged sites can be redeveloped relatively quickly, though some can persist for decades and some evidence of damage may be deliberately retained for a range of reasons. Damaged structures may remain for decades or centuries, especially if built on a large scale, and, if surviving, may undergo re-evaluation and retention as heritage features. This paper explores the fate of a range of relict features from the Second World War, surviving into modern urban and rural landscapes through inaction or deliberate action. Using a wide range of examples particularly from the UK but also drawing on others from elsewhere in Europe, we explore the impact of conflict on such relics; their nature and scale, processes of decision-making affecting their treatment over the last seven decades, and their landscape impact. The physical legacy of this war still affects many communities. Changing values so long after the conflict, and the decay of unmaintained structures, gives an urgency to reviewing the future of surviving relics.

**Keywords:** post-war reconstruction; relic features; urban landscapes; heritage; Europe

## 1. Introduction

The aftermath of most conflicts, at both small and large scale, has resulted in the reconstruction of cultural heritage. In some cases, this is reinstatement, whether restoration or replication as the victors of conflict impose new interpretations of heritage; in others, it is a form of new 'heritage'. Conflict produces 'dissonant heritage' [1], painful to most, hence the physical forms of that heritage are problematic. Perhaps because of that dissonance, the passage of time changes values and perceptions, and that heritage is re-evaluated. Whether these decisions are in any sense 'strategic replanning', a product of a wider-scale decision-making process, or are small-scale and local, varies enormously. This paper explores such reconstruction as product and process over the near eight decades since the end of the Second World War.

During the twentieth century more than at any previous time, warfare began to have a direct impact upon towns and cities, bringing 'civilians' into the firing line of conflict. Where ground fighting has directly affected cities, as with Ieper/Ypres during the First World War, the damage has usually been widespread and severe. However, it is with the long-range bombardment, by guns and particularly aircraft, that brought damage to cities far from the ground conflict. Although the damage was usually much more limited and diffuse, it created a new psychological impact. This was seen to greater effect in the attack on Guernica during the Spanish Civil War, and the concept of 'total war', involving all civilians of states at war, emerged [2,3]. During the Second World War, technology change allowed for more, larger and wider-ranging air raids, and the first cruise missiles and unguided ballistic missiles were used. Cities became major targets, both because they were

sites of military production, and for morale reasons. The latter led to the 'Baedeker raids' on English historic cities in 1942 [4].

"[Hitler] warns us solemnly that if we go on smashing up the German cities, his war factories and his bases, he will retaliate against our cathedrals and historic monuments . . . We have heard his threats before. Eighteen months ago, in September 1940, when he thought he had an overwhelming air force at his command, he declared that he would rub out—that was his actual expression, *rub out*!—our towns and cities. We have a long list of German cities in which all the vital industries of the German war machine are established. All these it will be our stern duty to deal with, as we have already dealt with Lübeck and Rostock and half a dozen important places . . . " [5].

The physical effects on many cities were immense, and direct and indirect effects have shaped cities—and wider landscapes—to the present day. This paper examines the impacts of this damage, identifying both strategic large-scale and local short-term planning issues, the impact of war-damaged structures in urban landscapes and the persistence of such relic features, and their treatment and re-evaluation today. These debates are explored primarily using UK examples, although many others elsewhere in Europe are mentioned, and the issues are of wide relevance, for example to war-damaged rural landscapes in eastern Europe.

This research follows a long-established historico-geographical tradition in urban research, where the built environment is primary evidence and decision-making processes are reconstructed through a variety of secondary sources [6]. Here, identifying surviving relict features draws on personal experience, site visit data and, to broaden the experience to continental Europe, the fast-growing sources of tourism, place promotion, social media and photography, the latter allowing online image searching. Secondary sources include national and municipal archives and published local histories. This multiple but necessarily unsystematic approach has been termed a 'scavenger approach' [7]. Digital sources and histories may be incomplete or erroneous, and multiple sources were cross-checked wherever possible. The work is further limited by the sporadic coverage of many areas, especially in eastern Europe and the understandable reluctance of communities to promote these neglected sites as places of memory or tourist destinations.

There is a large amount of literature on ruins as landscape features, and more on memorials including ruins as memorials: however, its main focus is on psychological reasons for retention and effects on users [8,9]. The scant literature on relict features in urban landscapes focuses on surviving and usually re-used, historic buildings [10]. This paper focuses on ruins as relict landscape features, and is exploratory research considering their identification, survival and future.

## 2. The Nature and Extent of Damage

Inevitably, damage varies according to the nature and severity of air raids, shelling and other destruction. High explosives are likely to demolish structures, leaving just rubble. Incendiaries may burn wooden structures but leave stone and brick in potentially repairable condition, although the severity of the firestorms in cities such as Hamburg and Tokyo left little remaining. The different conditions of war, construction materials and traditional architecture in varying countries affected damage (Table 1).

**Table 1.** Comparison of bomb damage in the Second World War [11].

| Damage | Britain | Italy | Germany | Japan |
|---|---|---|---|---|
| Civilian deaths | 60,595 | 59,796 | *c.* 600,000 | >900,000 |
| Cities suffering significant raids | *c.* 45 | *c.* 50 | 70 | 62 |
| Area destroyed (km$^2$) | *c.* 15 | *c.* 100 | 333 | 425 |
| Proportion of built-up area destroyed | 3% | *c.* 25% | 39% | *c.* 50% |
| Housing units destroyed | tens of thousands | tens of thousands | 2,164,800 | 2,500,000 |

This variation is also found within a single country and can be complicated by the way in which damage was recorded both locally and nationally (Table 2). In many UK cities, the raid-by-raid damage was charted accurately on Ordnance Survey maps, although the ways in which the severity of damage was defined is less clear. As damage was a matter of national security, it was classified, and some of these maps are still not readily available. London's maps are exceptional and have been published in atlas form [12]; Bath's maps indicate widespread small-scale damage, removing most of the city's Georgian window glass [13], while Birmingham has a more recent mapping of known bomb locations [14]. In many cities, unexploded bombs are still regularly found [15,16].

**Table 2.** Sample of city-level damage in the UK during the main 'Blitz' period [17].

| Town | Tonnage of High Explosive, 9/1940–5/1941 | Major Attacks (over 100 Aircraft), 9/1940–5/1941 | Acres of War Damage | Number of Houses Destroyed |
|---|---|---|---|---|
| London (County) | 18,291 | 71 | 1312 * | 47,314 |
| Liverpool/Birkenhead | 1957 | 8 | 284 | 7386 |
| Birmingham | 1852 | 8 | ? | 5065 |
| Plymouth/Devonport | 1228 | 8 | 193 415 * | 3593 |
| Glasgow/Clydeside | 1329 | 5 | ? | ? |
| Portsmouth | 687 | 3 | 182 430 * | 4393 |
| Hull | 593 | 3 | 136 246 * | 3324 |
| Coventry | 818 | 2 | 274 * | 4185 |
| Bath + | 400 HE bombs | 2 major raids but under 100 aircraft | ? | 1214 |
| Exeter + | 220 HE bombs | 1 major raid but under 100 aircraft | 40 75 * | 1700 |
| Norwich + | ? | small raids | 78 41 * | 117 |

Notes: +—Historic towns targeted by smaller 'Baedeker raids' in 1942. *—area of 'Declaratory Order' for reconstruction powers under the 1944 Act; but often includes areas of slum clearance as well as bomb damage. ?—No data.

Damage statistics may have been exaggerated or reduced for a range of reasons, for example relating to military security or political expediency. For example:

"We were told that 800,000 houses in London had been severely damaged. I am inclined to think that that figure is somewhat exaggerated. When the Prime Minister gave the first figures, and said that 1,000,000 houses were damaged, I felt that his advisers were giving him figures which, for some reason—I know not what—were greatly exaggerated. I felt that of that number, 75 per cent. or less had only tiles off or windows out, and that the actual figure of houses which require a lot of work is much smaller" [18].

The literature of the time, and even some more recent local and architectural histories, frequently use terms such as "totally destroyed", when incendiaries have burned the roof and interior fittings, but walls remain apparently little-damaged. A report on a lecture on London's bombed churches by Edward Yates FSA, for example, noted that some had been "completely destroyed except for walls and in some cases steeples" [19]. Notwithstanding such problems of accuracy, the work of reconstruction was immense: even by 1949 " . . . there is enough work of repair, rebuilding and replanning to last for the next two generations" [20].

### 3. The Initial (Emergency) Response

Models of the post-catastrophe reconstruction process show a stage of initial response. In the 1940s, after virtually each raid, this involved some stabilisation of ruined structures, the clearance of rubble, often closely followed by the demolition and clearance of unsafe structures. In much of the UK, the initial work was carried out by military personnel [21,22], followed by local authority workers and building contractors. However, in the haste to act and given that all such workers were unused to such circumstances, there were complaints that sound structures, or those that could have been repaired, were being cleared too speedily; this particularly applied to structures of architectural or historical value such as churches and public buildings. There are many mentions in *The Times* in early 1941, for example, of "unsafe buildings" being dynamited by the Royal Engineers and Pioneer Corps [23]. Thus, structures that survived the explosion or fire were subsequently, in many cases, lost. The influential magazine *Country Life*, for example, noted that "in too many cases, too zealous demolition has completed what the bombs only began . . . the arcades of All Hallows, Barking, which was damaged by fire, have now been pulled down" [24]. Action was taken by authorities such as the City of London, the Church of England and the Ancient Monuments Branch of the Ministry of Works to survey and safeguard churches and monuments [25].

In the UK, one response to this, most particularly for churches, was to identify individuals with relevant architectural expertise, to minimise post-bombing damage and clearance, and thus to retain bombed ruins for future consideration. These actions, particularly for bombed churches, fuelled developing ideas of conservation [26].

The location of cleared sites is likely to have affected subsequent development aspirations, such as the route of Birmingham's inner ring road, planned during the war [27]. Other bomb sites were used for functions such as car parking, with many becoming the basis for the post-war company NCP (National Car Parks).

*Emergency Development*

A second response stage is that of emergency development. This involves use of the cleared sites—or other locations—to provide some form of quickly-constructed replacement facilities. This allows for some level of continued community functioning and service provision.

In the UK, various forms of temporary shops were constructed in bombed cities such as Hull and Exeter. These were usually single-storey and often based on readily available temporary military structures such as Nissen huts [28]. Immediately after the war, the surplus capacity of many aircraft factories was used to construct temporary bungalows, widely known as 'prefabs', which were allocated to cities suffering particular housing damage. They used non-traditional design and materials with short design lives (5–10 years) and were located on all forms of undeveloped land: Birmingham was allocated about 6000 (Figure 1). Although such structures were designed and constructed as an emergency response, and with a short, planned life, for various reasons some have persisted for decades. Their very existence 'fossilises' an emergency response for an indeterminate, sometimes surprisingly lengthy, period, which may in turn constrain subsequent planning and development efforts.

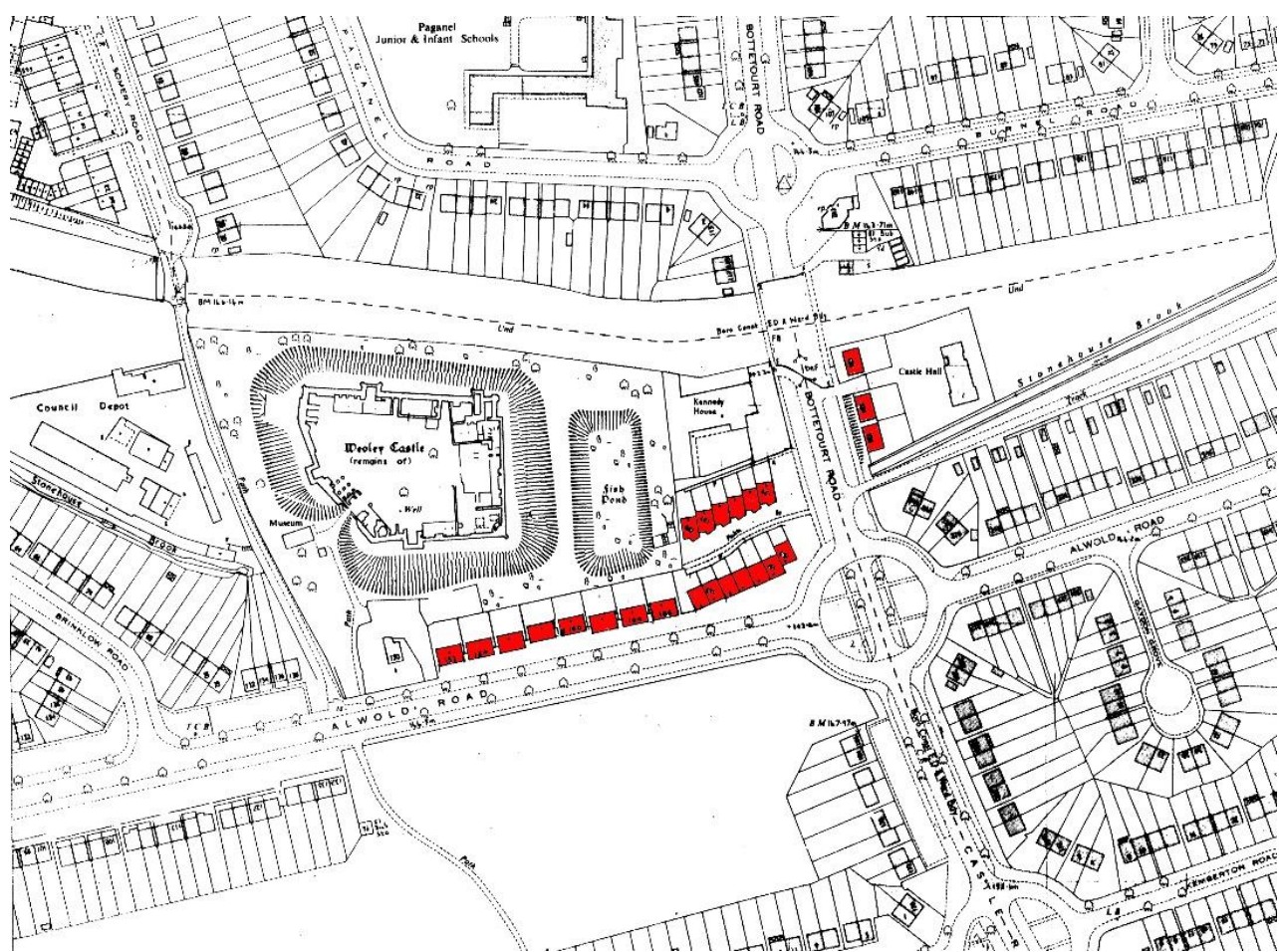

**Figure 1.** Locating 'prefab' bungalows (in red) around a medieval historic monument, Birmingham. Map of 1950s © Crown Copyright and Landmark Information Group Limited (2022). All rights reserved.

## 4. Planning and Reconstruction: The Permanent Solution

Reconstruction plans across the UK generally reached far in time and space, covering much more than the war-damaged sectors; the blurring of the line between 'blitz' and 'blight' was pervasive [29] and, half a century later, is difficult to disentangle. Professional town-planners formed a relative minority among the plan-makers: after all the profession was 'new'. The key agents in the conception *and* realization of the majority of reconstruction plans appear to be the City Engineer/Surveyor and the City Architect, although some—generally the best-known—plans were produced primarily by external and expensive consultants [30].

Many of the pre-1947 plans were highly illustrated and quite specific in their physical planning proposals. They were often visually appealing and effective at communicating ideas of the future city [31,32], but lacked implementation powers. After the Town and Country Planning Act of 1947, which introduced radical new planning mechanisms, the situation was reversed. Although many of the general ideas underlying the reconstruction plans, and the specific ideas contained within them, date back to before 1940, the post-1947 Development Plans were poor at communicating with a wider, non-professional, readership: reconstruction in this sense was not a 'new' paradigm.

While a varying amount of reconstruction planning took place during and immediately after the war, the reconstruction itself in all the blitzed cities took place following the passage of the 1947 Town and Country Planning Act. In many cases, it was some time afterwards, given the post-war financial crises and the continuing rationing of building material until

the mid-1950s [33]. Some bomb sites persisted for years, even decades, in places; they shaped experiences and became a familiar part of postwar culture [34–37]. Even ruined structures were familiar, both during as well after the war; and a new appreciation of ruins, as part of everyday urban experience, arose [8,38–40]. "Even though a ruin to-day is as common a feature of the street scene as a pillar-box, it still has this power to stir the heart. Even though we live and work among ruins, they still possess the beauty of strangeness" [41]. Yet the bomb damage provided the 'opportunity', as many professionals phrased it [42], to implement some of these older ideas for redeveloping the British inner cities and put in place the framework for the more substantial transformations that occurred during subsequent decades. These plans provided the modern urban landscapes we see and experience today, and were predicated on clearing damage, building on bombed sites, and perhaps deliberately erasing the memory of war, as was the case after the First World War in France [43] and after the Second, including levelling potentially repairable structures [44]. Memory and memorialising conflict is problematic in most contexts [1,45,46].

## 5. The Survival of Ruins

Ruins can be powerful social symbols that can carry a certain fetishization. This is particularly the case when the ruins are produced through the action of war, as the ruins may evoke memories of the place or building in its original state, of activities that took place there, and of people killed in the same conflict. The 'dissonance' in the heritage of war ruins and memorials arises between conqueror/victor and conquered populations, and when the distance of time has removed those most closely associated with loss and destruction. New residents and users have different values and memories. Yet memory is only a specialist theme in what appears to be a wider historical and societal fascination for ruins. Rose Macauley began her well-known book *The pleasure of ruins,* written in the immediate post-war years, by saying "to be fascinated by ruins has always, it would seem, been a human tendency . . . " [8], and this also permeates Woodward's more recent, more personal, volume [47].

As one letter-writer to *Country Life* put it even in 1944, "a state of ruin is in itself no bar to a beautiful existence" [48]. Macauley builds on this concept and includes a brief postscript 'Note on new ruins'. She deals explicitly with the ruins of war, including churches, and how they could, if allowed to weather, take on the same patina of age and familiarity about which she enthuses.

"Shells of churches [will] gape emptily; over broken altars the small yellow dandelions make their pattern. All this will presently be; but at first there is only the ruin; a mass of torn, charred prayer books strew the stone floor; the statues, tumbled from their niches, have broken in pieces; rafters and rubble pile knee-deep. But often the ruin has put on, in its catastrophic tipsy chaos, a bizarre new charm" [8].

The way in which these ruins could be perceived as tragic objects imbued with a certain aesthetic and/or symbolic beauty was highlighted even during the London blitz by the art historian Kenneth Clarke, who suggested that bomb damage was in itself Picturesque (quoted in Woodward, p. 212 [47]). Although the idea that such damage could be seen in the same way as the great English contribution to landscape philosophy was challenging, such a perspective became popular amongst some architectural writers at least.

The idea that some, at least, of the bombed churches in London and elsewhere might be retained as ruins, and used as public open spaces, gardens and war memorials, was raised soon after the main blitz. Sir Edwin Lutyens, probably England's best-known architect of the time, wrote to the architect S.A. Alexander on 16 January 1941 that, despite the need for space for housing, "where there is no congregation I would leave the spaces occupied by destroyed Churches as open" [49]. Shortly afterwards it was suggested that " . . . certain churches could effectively remain as ruins . . . [not the too-burned Wren ones, but—for example—Hawksmoor's St George's in the East; Archer's St John Smith Square] . . . If it is not wanted as a place of worship why not let it remain as a shell, a witness—and a beautiful one—of the acts of these times as well as of its own" [50]. In their iconic reconstruction

plan for Plymouth, Watson and Abercrombie suggested that the ruined Charles Church should be retained as a ruin, "a fitting memorial to symbolize the city's grief and honour in the triumphant survival of the trials of this tragic war" [51].

Probably the best-known advocacy of this idea was through the publication in 1945 of a slim book, *Bombed churches as war memorials* (Figure 2) [52]. This built upon an earlier article by Jellicoe and Conder for the *Architectural Review*, using many of the same illustrations [53]. It was introduced by Hugh Casson, and contained detailed proposals for Christ Church, Newgate; and the adjoining ruins of St Alban, Wood Street, and St Mary, Aldermanbury, in the City of London; and St Anne, Soho, and St John, Red Lion Square, elsewhere in London. This book was well illustrated with sketches and even detailed planting diagrams. Casson argued strongly against the purely functional and financial arguments that these churches had largely lost their congregations, and their valuable sites would raise money needed by the Church elsewhere.

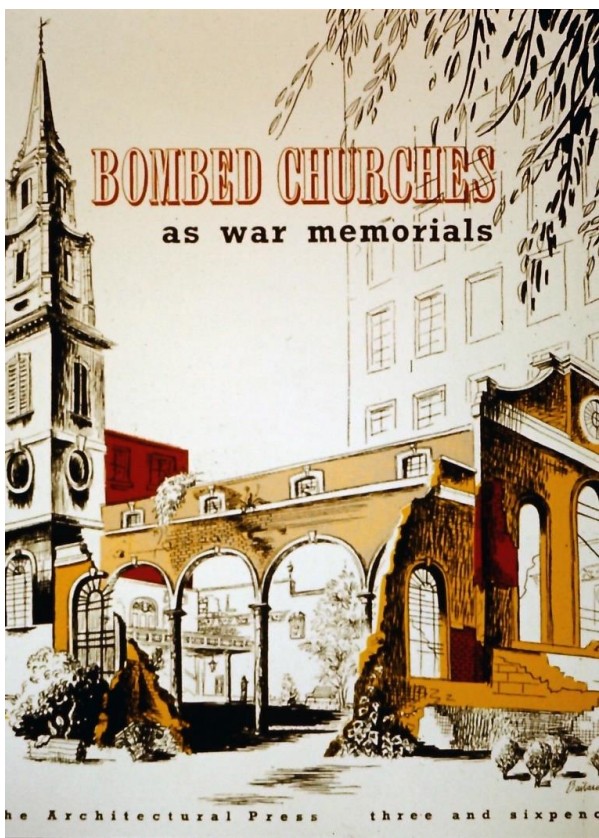

**Figure 2.** Strikingly illustrated cover of *Bombed churches as war memorials* (authors' collection).

The experience of war and destruction led to a range of international activity including charters such as the 1954 Convention for the Protection of Cultural Property in the Event of Armed Conflict (the 'Hague Convention'). However, these seem to have had more immediate effect on decision-making on the European mainland [54]. Unless an area has an international designation such as a World Heritage Site, our experience in the UK is that the direct effect of such charters is extremely limited. This burst of international activity [55] seems to have had greatest effect in ensuring that heritage and conservation issues are central to processes of development and spatial planning [56].

Nearly eighty years after the end of the war, and after several cycles of large-scale urban redevelopment, most of the bombsites and ruins have been removed, in urban Western Europe at least. Those who study bomb damage maps can still identify a small number of open sites, never redeveloped. Otherwise, most memorialising of war damage has been reduced to shrapnel damage (or, still in Berlin, artillery shell and bullet damage)

and plaques, for example that in London marking the location of the first bomb dropped on the city on 25 August 1940.

There are a few rare survivors, though. In both Hull and Bath, individual bombed buildings have survived to the present day as ruins. A bombed cinema in Hull, long neglected but given State protection as an historic monument for its rarity as a surviving bomb site, is about to be converted to a civilian war memorial, with a grant from the National Lottery Heritage Fund [57,58]. In Bath, a bombed but patched-up municipal office building, also now protected, has retained its scars in a major rebuilding [59]. However, both of these structures are located outside the busiest and highest-value business and commercial areas.

More common are the ruins or sites of bombed churches that have been deliberately retained, often with some form of memorial function [60]. These are 'relict features' in contemporary urban landscapes, although they are not unique: in England at least, there are significant numbers of other remnants of disused or ruinous churches remaining in both urban and rural landscapes, with there seeming to be some reluctance to completely dispose of the relics of buildings once used for religious purposes. Ongoing exploratory investigation has identified hundreds of such war-relic religious buildings across Europe (Figure 3). This work is limited by the authors' linguistic ability and the paucity of information, especially online, and especially relating to the shifting borders of the former Eastern Europe: it is likely that more detailed local investigation will identify further examples in this zone. Owing to the political circumstances of the post-war period, with forced movement of communities, military control of border zones, and poor rural economies, some of these relics are not direct war casualties by bombing or gunfire (as is the case with UK and most Western European examples). However, they are still indirectly casualties of war and their management and use and/or potential use in terms of modern planning merits consideration.

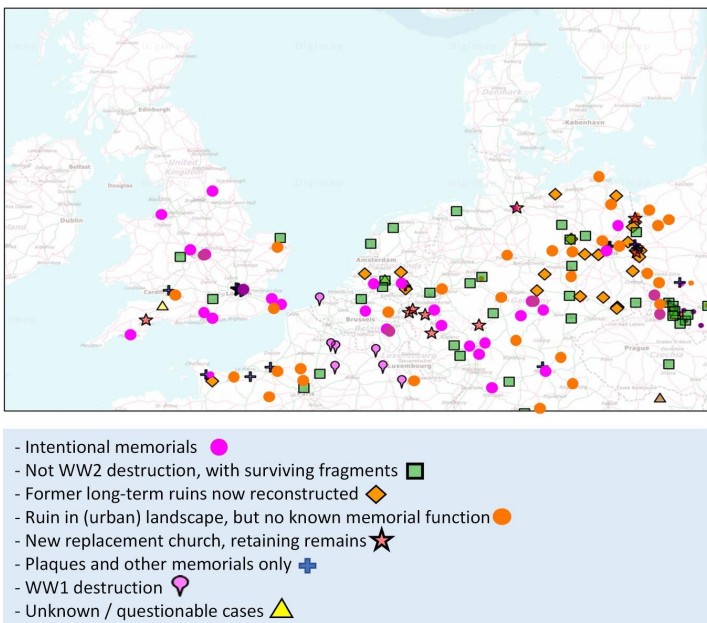

**Figure 3.** Remaining churches partly destroyed or abandoned following the Second World War (from authors' research to June 2022 using site visits (in the UK), archives, local histories, tourist information and digital image searches to identify war-damaged churches, located on a map base using Arc-GIS).

Many of these examples have complex stories of decision-making leading to retention, preservation, or at least a degree of ambivalence to demolition and rebuilding [61]. Figure 3 shows an interesting balance between those ruins having an intentional memorial function, of which there is a relatively high concentration in England, and those with no known intentional memorial function. The retention of Coventry's bombed cathedral as an adjunct

to the eventual competition-winning design for its replacement is well known; less so is the earlier scheme to incorporate some of the ruined structure into a replacement [62]. The retained church in Plymouth is particularly prominent as it now stands isolated on a roundabout of a high-speed inner dual carriageway, backed by the strikingly designed outer wall of a shopping centre; not what was envisaged in the iconic *beaux-arts* city-centre plan which clearly shows the Charles Church, surrounded by its churchyard, adjoining the road not within the traffic island [51].

This example highlights that many churches are prominent landscape features, whether for their location, scale (especially height of towers or spires), materials and styles. Those retained as urban ruins often retain such visible landmark status in addition to newly acquired memorial status, while those in rural communities were often, and usually still remain, the most prominent public structure in those areas (Figure 4).

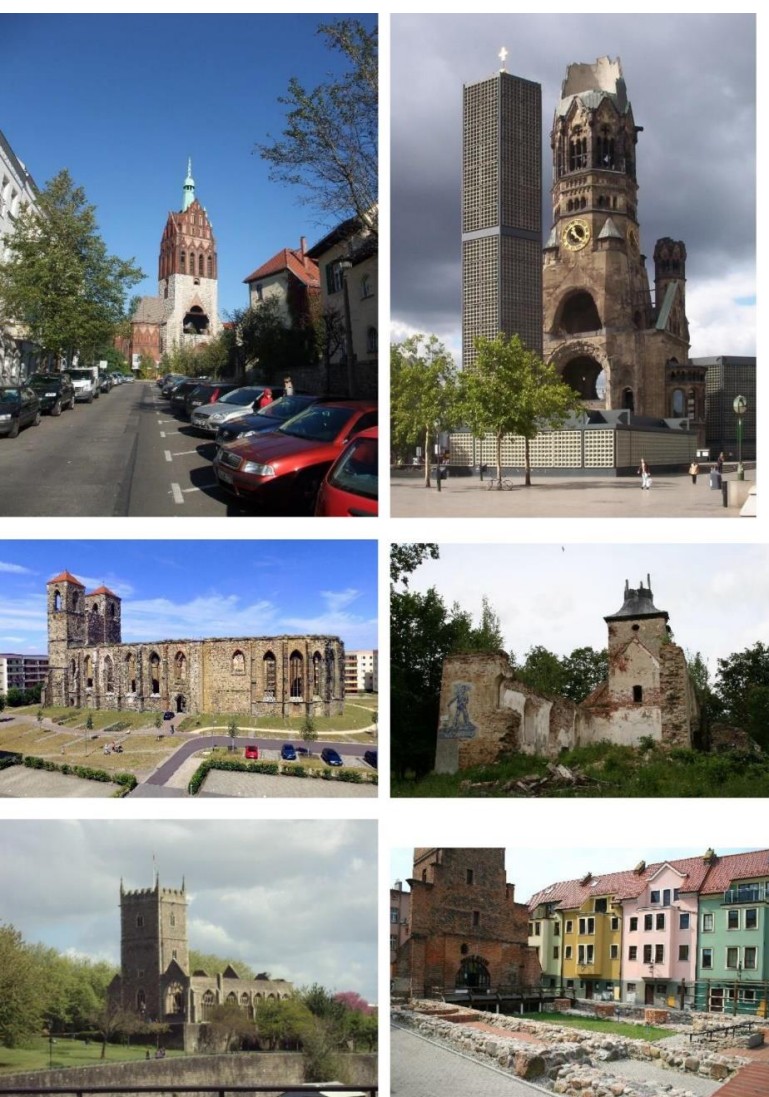

**Figure 4.** War-damaged churches. **Top left**: Bethanienkirche, Mirbachplatz, Berlin: a prominent location (photograph by Sebastian Wallroth, CC-BY-SA-4.0). **Top right**: Kaiser-Wilhelm-Gedächtniskirche, Berlin-Charlottenburg, a national memorial function (photograph by Null8fuffzehn, German Wikipedia). **Centre left**: St Nikolai, Zerbst, ruin in the centre of a rebuilt residential area (photograph by Bodow, CC-BY-SA-4.0). **Centre right**: rural ruin of Borchersdorf-Selenopolje, Kaliningrad Oblast (photograph by Kno-Biesdorf, CC-BY-SA-4.0). **Bottom left**: St Peter, Bristol, ruin in a new urban city-centre park (photograph by Rept0n1x, CC-BY-SA-3.0). **Bottom right**, ruins of St Catharine's church, Bytow, surrounded by historicist reconstruction (photograph by Danapass CC-BY-SA-3.0-PL).

## 6. Political, Ideological and Other Influences

Whatever the models of reconstruction may suggest, the reality is often rather different. While the urban war damage in much of Western Europe had largely been rebuilt by the time of the next crisis—the 1973 Middle East war and oil shortage that suddenly halted much building activity—that was not the case in much of Eastern Europe. Some of this delay was political, ideological, or resource-driven (though resource allocation is often also political). As Cochrane notes in relation to Berlin, decisions about destruction, monument retention and even monument creation revolve around power, the manipulation of memory and the creation of identity [63]. Such control decisions can create further heritage dissonance. The decision to demolish the ruins of Berlin's Stadtschloss by the DDR government in 1950–1951 was overtly political, wiping away a symbol of the old Germany and, a couple of decades later, constructing the Palast der Republik [64,65]. Small rural towns such as Szydłów (Poland) were only being rebuilt in the late 1980s, while in 2000 Pawłowski had considerable problems "to make my Western colleagues realise that the issue of totally destroyed towns has still remained a current issue in some parts of our continent" [66]. In Eblag (Poland), for example, "after forming a public park for 20 years, the buildings of the old town are now being reconstructed on their old foundations, approximately to their new heights, but in a rather frenetic post-modern style. This whole rebuilding process has been dubbed 'retroversion'" [67,68].

Dresden's Frauenkirche, reduced to a pile of blackened rubble following the post-firestorm clearance, was deliberately retained in that state "to symbolise the barbarism of the Allies" [69]. The bombed historic district around it was redeveloped with modernist, boxy structures. It remained in this state until the fall of the communist regime. Soon thereafter, a $156m project to recreate the church was undertaken, with multiple purposes relating to wartime victimhood and 'undoing trauma', the disappearance of the communist regime, and a new national identity [69].

Warsaw is often cited as an exemplar of historicist reconstruction after catastrophe. However, some of the decision-making, including 'inaccurate' interior rebuilding to give better living conditions, mirrored inter-war urban improvements elsewhere in Europe and, likewise, began before the conflict [70]. The severity of wartime destruction and the need to reinstate a national capital, under a wholly different political regime, led to a new outlook: the interventions "sought to eliminate 19th-century capitalist development", especially of Germanic influence [70]. However, the emerging ideology of 'preservation' and reconstruction was driven by cultural and national identity, of which the damaged monument was a critical symbol [71]. Both the approach to Warsaw's monuments, and the neglect of others (including war-damaged churches) elsewhere in communist Poland, arose from "the fact that decisions about historic buildings during the Communist period were made with the ideologically charged malevolence as a predominant factor" [72].

On a much smaller scale, in some bombed German towns there was a greater concern for 'authenticity', of not replicating what had been destroyed, but seeking new building structures that would give a contemporary impression of what had been lost. In Nuremburg, for example, the destroyed tightly packed traditional steep-roofed timber-framed structures were largely replaced by modern structures which, though un-ornamented, had steep pitched roofs and balconies in an attempt, as the chief planner Heinz Schmeissner suggested, to secure "the preservation of the concept of Nuremburg". The survival of extensive remains of historic structures influenced this decision [73].

Some of the ideological influences concerned the nature of repair, retention or restoration. The contribution of the bombed churches and schemes for repair or retention as ruins has been mentioned. In the UK, the replication of destroyed fabric is exceptionally rare—there are examples in London and Bath, but this was usually restricted to repair of historic terraces. This can be partly attributed to the professional acceptance of the views of a campaigning conservation organisation, the Society for the Protection of Ancient Buildings (SPAB) [74] although, by the end of the 1930s, SPAB "remained set in the distant past" [75].

This approach markedly contrasts with examples on mainland Europe, in Ieper/Ypres after the First World War and Warsaw after the Second, for example.

One important UK example is Norfolk Crescent, Bath (begun 1792). The northern part of the terrace was gutted, and about six bays of the façade had been lost. The façade was reinstated in 1958 by E. F. Tew and the structure converted to local authority-managed flats, although the rebuilt section is deeper than the original and "the rear does not attempt a Georgian reconstruction" [76]. The conversion of this and many other terraces followed SPAB advice by the architect Marshall Sisson and others [77]. A Civic Trust Award plaque now commemorates the scheme. The undamaged part of the terrace was Listed Grade II* in June 1950 (i.e., the second-highest grade of State protection for historic structures, a system called 'Listing', the structures being 'Listed Buildings') [78]. In Queen Square, the Francis Hotel's east wing was badly damaged. Again, although the façade was rebuilt, by J. Hopwood in 1952–1953, to match its neighbours, the new wing is deeper in section. "As a component part of an outstanding set-piece, meticulously reconstructed in facsimile in 1955 (see inscription on front), these houses remain of great importance" and are part of the Grade I listing of June 1950 (i.e., even before reconstruction) [79]. The two seem to have been treated similarly in terms of reconstruction authenticity, where the public façades had greater significance, yet differently for listing (heritage/preservation) purposes—although Queen Square is undoubtedly a more significant architectural composition. The significance of the Norfolk Crescent example is that it was mentioned with approval in an influential 1960s book on conservation by Roy Worskett, sometime chief planning officer for Bath [80]. Such 'semi-replica' terrace restorations in Bath, Leamington Spa and elsewhere were inauthentic in plan, but suitable in façade. They responded to a growing 'townscape' philosophy in British post-war planning.

A more substantial case is Park Crescent, London. This was an 1820s development by John Nash, part of a major new urban landscape from Regent's Park to St James's. Part of the crescent was badly damaged and, though given the highest grade of State protection (a Grade I Listed Building), the damaged remains were demolished and a partial replica rebuilt in 1962. The new structure retained its Listed status, but because of its value as part of the planned townscape rather than any intrinsic architectural merit. By the early 2010s this replica was felt to have functional and structural problems and demolition and construction of a more accurate replica façade was proposed [81]. This leads to consideration of the reappraisal of reconstruction-era decisions and actions, seven decades or so later.

## 7. Reappraisal

This is a stage absent from the usual models of post-catastrophe reconstruction, but is something that should be considered some decades after the initial reconstruction as buildings and areas age and societal expectations change. There are two issues to consider here. First, that reconstructions, however inauthentic, can be perceived by subsequent generations after the passage of some time as 'genuine monuments' [82]; and secondly, as relict structures decay, their memorial value can alter as those directly affected by the catastrophe which they commemorate die and relatives or incoming residents and users have other priorities. The first point is well-demonstrated by efforts to secure World Heritage status for places destroyed and rebuilt after the Second World War. ICOMOS and UNESCO increasingly considered issues of 'authenticity'. Hence, the north side of the Markt in Weimar, rebuilt 1988–1993 after wartime damage to recreate the character and enclosure of the pre-war market place, is not included within the World Heritage Site. The serial rejection of rebuilt Gdańsk for World Heritage status in the late 1990s has been attributed to the "antiquated stance" of one influential individual "concerning the authenticity of fabric as the prerequisite of inscription" [83]. Perhaps post-catastrophe reconstructions should be considered as a distinct class of heritage, requiring different treatment [84,85].

The second point is illustrated by the remaining bombed churches in England, most of which were disposed of by the Church of England and other denominations fairly soon after the war and are now the responsibility of local authorities. Maintenance of such structures is expensive and funding, of course, is lacking, particularly since the 2008 economic crisis. The continued conservation of these structures may be increasingly doubtful, given their low apparent use and the length of time that has elapsed since their ruination. Families that may have lost members in these calamities may well have moved away or moved on; incoming residents have little or no special attachment to these places. Despite this lack of direct engagement with these sites, any suggestion of redevelopment is likely to be contested. In this sense, shifting socio-economic, political, environmental and physical conditions affect building 'redundancy'; while personal tastes, trends and fashions can all influence debates around adaptability.

Many of the remaining damaged or abandoned churches in Eastern Europe continue to decay, as a number of recent documentary photography and social media projects demonstrate [86,87]. Political changes have not, on the whole, led to increased rural prosperity to afford reconstruction, and original religious communities have not moved back. On the other hand, there are growing examples of communities, or even individuals and families, undertaking projects that might more accurately be termed repair rather than accurate restoration. An example is the church of Aruküla, Estonia, being restored largely by one individual [88].

The urban decay and changed priorities in rebuilt Marseille allowed for an archaeological investigation of the destruction/reconstruction sites. This is a perhaps extreme example that has allowed for a reinterpretation of both the destruction and the reconstruction [44].

In the case of London's Park Crescent, mentioned above, the reappraisal came about because of problems with the post-war structure. Although this was recognised as "an interesting and unusual example of post-war reconstruction of a 19th century terrace" it was agreed by Historic England, the State heritage advisory body, and the Mayor of London as the planning authority, that:

"The proposed redevelopment would remedy these deficiencies and result in a crescent that is far more historically and architecturally authentic when viewed from the public realm. The reinstating of the original complement of front doors and the chimney stacks is particularly welcome . . . the replacement building would both sustain and enhance the significance of this internationally important townscape, making a significantly greater positive contribution to the character and distinctiveness of the Regent's Park Conservation Area and resulting in a high-quality development that is sensitive to its heritage context" [81].

The replacement structure was completed by 2019 (Figure 5), overcoming many technical problems inherent in creating a replica to modern structural and environmental standards but retaining the Grade I Listing [89]. This example demonstrates an unusual approach to war damage: the replication of an original structure, though to a low grade of accuracy; and its own demolition and replacement as a result of more recent reappraisal. This reappraisal is becoming more common, indeed necessary, for relict structures at this distance from the cause of their damage, and in such different cultural, financial and economic circumstances.

Although particularly prominent, this approach is not unique in this area of London: the driving forces of property values and heritage are such that several other terraces have been substantially rebuilt behind retained façades, and the patched-up and Grade I Listed Cambridge Terrace was rebuilt in facsimile in 1986 "restoring exact external details and symmetry of the terrace" [90].

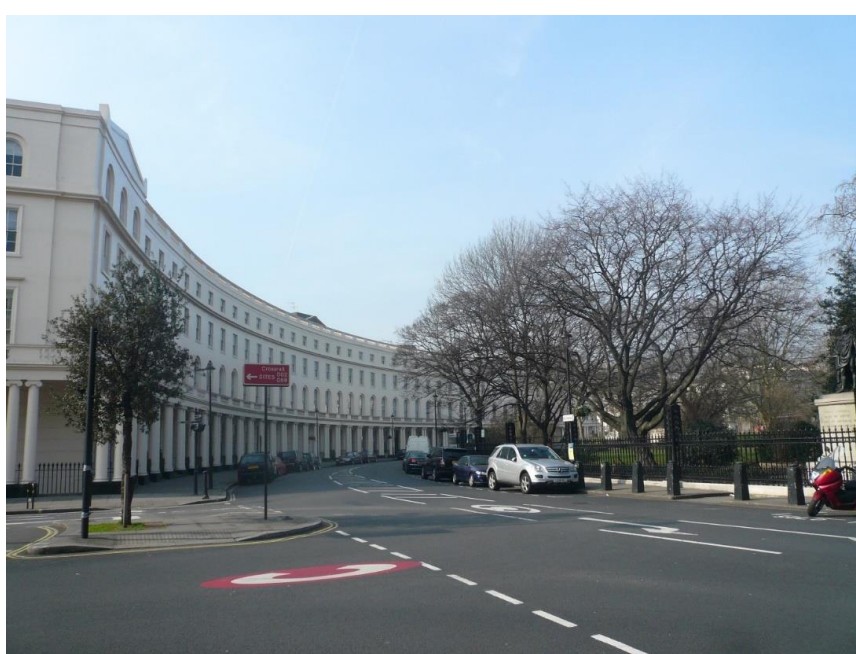

**Figure 5.** Park Crescent, London: redevelopment of post-war replica. The visually identical but more original part of the crescent is behind the photographer (photograph by P.J. Larkham, 2019).

The unanticipated survival of some temporary structures—prefabs and shops—far beyond their initial anticipated lifespan has also been problematic. While there has been a social and historical reappraisal of the prefab bungalow communities in the UK [91,92], the fabric of many surviving structures has been difficult to bring up to modern living standards and, in the case of many of the survivors, the buildings have been essentially rebuilt on their original foundations and to a similar scale, as is the case with a prefab development in Wolverhampton. Some other survivors, such as a line of 17 in Birmingham, of a rare design, have received protection through Listing (Figure 6). Temporary shops in particular, occupying more valuable commercial sites, have largely vanished although some, such as a group in Exeter, persisted until a major urban redevelopment in the mid 2000s [93].

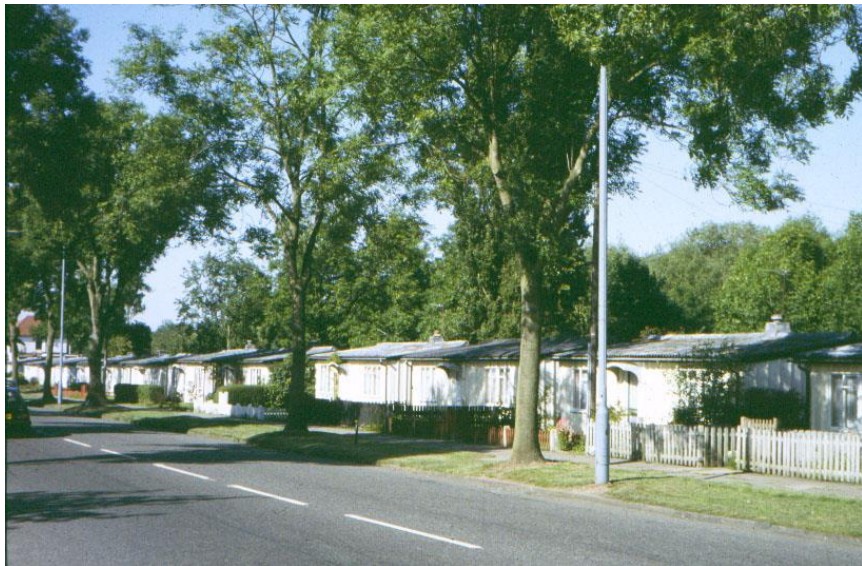

**Figure 6.** Listed 1946 prefabs, Birmingham (photograph: P.J. Larkham).

Bristol provides a further example where reappraisal of the first generation of post-war reconstruction is leading to radically new approaches. The city core was badly bombed and, while a new retail centre was developed on a new site, three ruined churches in close proximity have been retained: one, St Peter, at the centre of a new public park, and the other two, including St Mary, at the edge of that park, having new commercial building wrapped closely around their remains. However, half a century and more since the reconstruction, fashions and tastes have changed. The 1950s offices around St Mary have become neglected, unfashionable and vacant; in fact, a graffiti-sprayed eyesore. So, a recent development proposal would remove these blocks, reinstate some of the medieval street alignments that were lost in the 1950s, making the St Mary's ruins more visible and accessible [94]. As water reputedly adds value to developments, so too, it would seem, does a ready-made landmark, even if it is a large-scale reminder of a difficult past. This new third-generation urban landscape would substantially change the perception and use of this bombed church. It is, though, difficult to conceptualise the two generations of post-war urban landscape in traditional morphological terms: plot patterns have gone, relationships between ordinary and special buildings have been changed, and street patterns have changed but may be reinstated. However, the scale of proposed new buildings, and their impact on protected structures including St Mary's, has caused concern [95].

South of the river in Bristol, and close to the third bombed church, another redevelopment proposal recreates a bombed urban block on a site that has been a surface car park since the war. The competition-winning design provides 120 apartments in a design "informed by buildings lost" in the bombing, "wrapped in a bronze mesh that replicates the scale and forms of previous 18th-century buildings that occupied the site" (Figure 7) [96]. Construction is expected to begin in 2024. Such reinterpretation seems more acceptable to the planning and conservation culture in the UK, rather than the near-facsimile reproductions that are becoming common, and causing some concern, elsewhere [97,98].

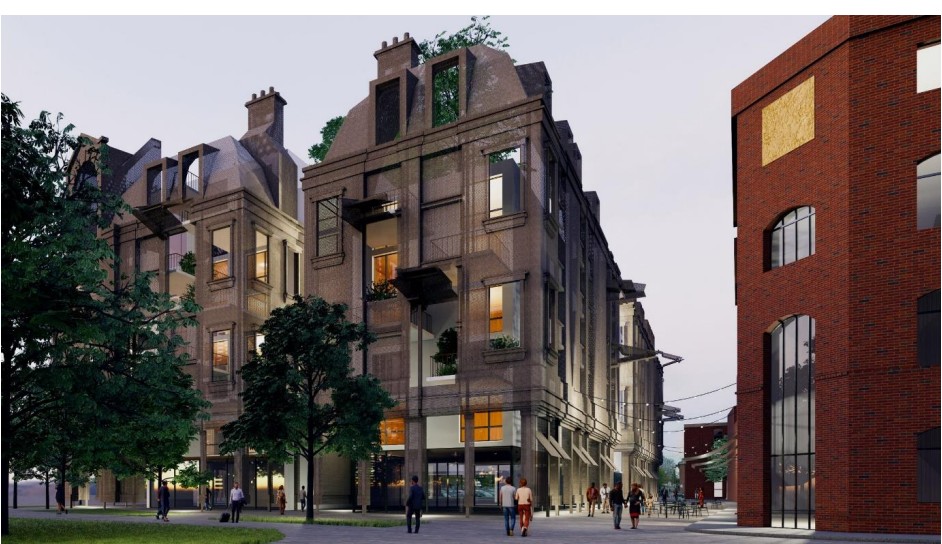

**Figure 7.** Architect's visualisation of the proposed Redcliffe Way redevelopment on a bombsite/car park, Bristol. © GROUPWORK, McGregor Coxall, Hydrock, reproduced with permission.

## 8. Conclusions

Catastrophe such as wartime destruction usually pays little heed to society's valued structures—indeed they may instead become specific targets, as with the Baedeker raids of 1942. Hence, historic monuments, buildings and areas are likely to suffer significant damage [99]. Replanning and rebuilding are often carried out speedily: a crisis needs a swift response and it is often difficult to appropriately consider how heritage materially contributes in reconstruction. The need to replace functioning buildings such as houses, shops and offices often means that any assessment of heritage 'value' remains a low priority.

The physical impact of war on the built fabric is, therefore, significant and long-term. There are economic, land-use and functional aspects; together with psychological impacts on residents and users, whether or not they are descendants of those killed or injured. This paper has identified a wide range of possible responses, which could broadly be categorised as:

- Retain site and rubble, for political/ideological/social/psychological reasons (often a relatively short-term option);
- Retain site and rubble because there is neither public or private will or resource to do anything else;
- Clear rubble and retain site as public space, with possible memorial function, or as commercial space, e.g., car park;
- Retain damaged structure and some or all of its site as public space, memorial or other function;
- Repair damaged structure to usable condition, potentially retaining some of the damage scars;
- Restore damaged structure to condition as before the damage;
- Clear site and redevelop entirely new structure;
- Clear wider area and redevelop entirely new urban structure.

After a period of years, temporary redevelopments are replaced or, if they persist, might become 'heritage', and hence woven into an authorised heritage discourse. Even the 'permanent' reconstruction will age and be re-evaluated; being either redeveloped, retained and refurbished, or retained as part of an authorised heritage discourse. Most commentators identify fewer options: the architect Lebbeus Woods, for example, while researching architectural responses to destruction in Sarajevo and Beirut, felt that there were only two, so he 'invented' a third: "the post-war city must re-create the new from the damaged old" [100]. Yet, surely, all cities must achieve this, unless the reconstruction decision is to move to a wholly new site (as was proposed for Hannover [101] and Lorient [102], for example).

While some of this post-conflict reconstruction might be seen as 'strategic', if conceptualised at regional and national scale, not every place had a regional or national plan. This may also partly explain the abandonment or neglect of so many war-damaged rural buildings, particularly in Eastern Europe. Despite various campaigns, the UK did not produce a post-war national plan, for example. Most city reconstruction plans were produced at city scale, or even for specific parts of cities. Although many had timescales of 20–50 years, this is only partly strategic; the wider spatial scale is also important. The likelihood is, therefore, that much post-catastrophe replanning could hardly be considered 'strategic' in vision or scope.

The implications of this work are widespread in both space and time. The decisions leading to the survival and subsequent treatment of relics of past conflict can inform post-conflict decision-making after current and future conflicts irrespective of their spatial location. This is the case in those cultural and administrative contexts where international charters such as the Hague Charter exert a more immediate influence than is evident with UK local authorities. These relics are relatively small in scale and, with the rare exceptions such as Coventry Cathedral, relevant decision-making needs to be at the local level, taking account of local views and values. This work can also inform decision-making following other forms of widespread destruction such as natural disaster, and climate change may be affecting the nature, scale and frequency of disasters. At the time, it is important to separate immediate, 'emergency' actions and constructions. A later phase of 'permanent' reconstruction is likely to require removal of these temporary structures, though this may itself be problematic. Finally, in the more distant future, the entire rebuilding will need reappraisal.

It is the reappraisal of the post-war reconstruction landscape that is a critical concern for present-day urban managers and strategic planners. Not only is this becoming a necessary task as rebuilt buildings and wider urban landscapes age, and contemporary values and requirements change; shifting towards the sustainability discourse and its

general emphasis on retain and reuse. This paper has set the UK war-damaged relict landscape features, about which a considerable amount is known, in the wider context of European landscapes where, particularly in Eastern Europe, the proliferation of such landscape features has been an unsuspected finding of this 'scavenger approach' research. There is scope for future research to, more systematically continue to identify these features, to examine their condition and use, and how changing attitudes towards the contested and dissonant heritage of war, destruction and catastrophe.

**Author Contributions:** Conceptualization, P.J.L. and D.A.; Methodology, P.J.L.; Investigation, P.J.L. and D.A.; Writing—Original Draft Preparation, P.J.L.; Writing—Review & Editing, D.A. and P.J.L.; Visualization, P.J.L. and D.A. All authors have read and agreed to the published version of the manuscript.

**Funding:** This research received no external funding.

**Conflicts of Interest:** The authors declare no conflict of interest.

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
