# Peer review of "Relics of War: Damaged Structures and Their Replacement or Management in Modern Landscapes"

_sustainability, doi:10.3390/su142013513_

Round 1
Reviewer 1 Report
- The review is clear enough to understand the scope of its focus. It has relevance to the field. Cultural heritage and protection of heritage areas is of iimportance. The paper tells the British experience about war relics and the potential uses of war damaged areas.
- I think this topic will get attention of more people especially culturally aware people on the globe. In fact sÅŸnce WWII the topic is hot.
- The authors correctly cited resources used and listed them in references. However it is shocking that they never referred to 1954 UN Convention for the Protection ov the Cultural Property in the event of Armed Conflict.
- There are several visuals used in the document and they are appropriate and easy to understand.
Reviewer 2 Report
The topic is very interesting and clear presented.
The methodology should be explained.
Pro-and-cons discussion is missing.
Also, the adopted international Charters and Agenda`s are not mentioned as well as the attitude in relation to them.
Reviewer 3 Report
The authors dealt with the problem of the relics of objects destroyed during World War II, which is interesting from the point of view of shaping the urban space and other spatial arrangements. Despite the passage of time, they have been preserved in different conditions, as shown in the summary of the article. However, the reviewed work is not a scientific article, there is no appropriate structure, and above all, the methodological part. The aim of the work has not been precisely specified and needs to be supplemented. There is no discussion of the methods by which the source material was collected and analyzed. I understand that this is a case study, not very representative of the relics of World War II, as most of them come from Great Britain. Most of the ruins after the war were in the territories occupied by Nazi Germany in Central Europe and the Balkans, although the authors mention the reconstruction of the completely destroyed Warsaw or Gdańsk in Poland, but these are few examples. In the historic city of Krakow, the German occupiers demolished, for example, the monument of King Władysław Jagiełło, the founder of the dynasty, victorious in the famous Battle of Grunwald, but the society of the city of Krakow rebuilt this monument immediately after the war, as did many others. Bearing in mind such a selection of examples for the analysis, it should be added both in the title and in the abstract that this applies to selected objects, mainly from Great Britain.
Reviewer 4 Report
Thank you for giving me an opportunity to review the paper entitled: “Relics of War: Damaged Structures and their Replacement or Management in Modern Urban Landscapes”.
There are several suggestions given for the improvement of the paper:
1. The abstract of the paper is well written. However, authors should also highlights the urgency of the study.
2. I found that the intention of the study is clearly explained and expressed.
3. However, author may need to provide some explanation why there is a need to conduct this study. Author should highlight and explain the urgency of the study. Why this study should be conducted? What lead author to conduct this research? Author should explain it thoroughly, especially the urgency of this research. Authors should also include and discuss several relevant studies. I wish to see more how this study differ from these past studies and talk about the novelty of the study.
4. I would suggest an implication section which focuses on the key insights could be offered to local government or nation, if relevant.
5. Any limitation of this study? Any recommendation for future studies?
Reviewer 5 Report
I would like to thank the authors for their contribution. The paper is generally well-written and -structured. This is why I have only minor comments. It is not clear what the significance of this study is in the current context. If it is repurposing these structures, what would be the benefit of it? Although the authors address it in the conclusion, the introduction should also justify the importance of this study. When the authors mentioned the concept "dissonant heritage" I thought it would later come up or be elaborated. However, it does not really explain how the repurposing of these structures could contribute to the dissonant heritage. Here, I suggest authors refer to publications outside the British literature sources.
The methodology is not really clear. If the authors conducted archival research along with document analysis, they should explain from which institutions the images, maps, and documents are obtained. Similarly, the figure 3 caption is not clear. It says that it is from the authors' research. Is there a broader project? If yes, the authors should state what that broader project is even if it is unpublished. Is the map created by using QGIS? How these sites were pinpointed? Where does the data come from? I suggest authors give precision to the figure.
There is no page number for me to be able to better explain but under the section "Political, Ideological and Other Influences" there is one sentence as a paragraph that is self-standing. It should be either further explained or integrated into other paragraphs. In some cases, there are double spaces, which need to be revised.
I hope that my comments are useful for authors to revise their paper for its publication.
